# Educational Intervention to Improve Citizen’s Healthcare Participation Perception in Rural Japanese Communities: A Pilot Study

**DOI:** 10.3390/ijerph18041782

**Published:** 2021-02-12

**Authors:** Ryuichi Ohta, Yoshinori Ryu, Jun Kitayuguchi, Chiaki Sano, Karen D. Könings

**Affiliations:** 1Community Care, Unnan City Hospital, Unnan 699-1221, Japan; yoshiyoshiryuryu.hpydys@gmail.com; 2Physical Education and Medicine Research Center Unnan, Unnan 699-1105, Japan; junk_907@yahoo.co.jp; 3Department of Community Medicine Management, Faculty of Medicine, Shimane University, Izumo 693-8501, Japan; sanochi@med.shimane-u.ac.jp; 4School of Health Professions Education, Faculty of Health, Medicine and Life Sciences, Maastricht University, Universiteitssingel 40, 6229 ER Maastricht, The Netherlands; kd.konings@maastrichtuniversity.nl

**Keywords:** citizen participation, educational workshop, healthcare, older rural citizen, social cognitive theory

## Abstract

In this mixed-methods study, we hypothesized that social cognitive theory (SCT)-based educational interventions for healthcare participation can improve the self-efficacy of older rural citizens in participating in their health management without any difficulties. Quasi-experimental study before and after SCT-based educational interventions and semi-structured interviews were conducted. Participants were Japanese elderly (>65 years) from rural communities. Propensity score matching was performed to estimate the effectiveness of educational interventions on participants’ perception (intervention: *n* = 156; control: *n* = 121). Interview contents were transcribed verbatim and analyzed based on thematic analysis. The intervention group scored significantly higher than the control group for participation in planning and managing self-care. Interviews revealed three themes: ability to manage health conditions, relationship with medical professionals, and relationship among citizens. Participants reported difficulties in judging symptoms and communicating with medical professionals. Hierarchy and low motivation to participate in healthcare hindered collaboration. The findings suggest that SCT-based educational interventions can positively impact rural citizens’ self-efficacy in healthcare participation.

## 1. Introduction

Ordinary citizens’ primary role in health control is to participate in their own healthcare by cooperating with families and healthcare professionals and to be actively involved in the decision-making process related to their own healthcare. Through such shared decision-making processes, patients can obtain vital information from healthcare professionals and discuss with them their preferred care methods [1], which can improve patients’ care satisfaction and sense of empowerment [2]. However, patients may be afraid of being labeled “difficult,” and hence defer to their physicians’ opinions even when these opinions contradict their own [3]. 

To improve patients’ participation in their health management, it is essential to improve their health literacy [4] and shared decision-making skills [5]. A patient’s age, culture, and environment affect their health literacy [6], and patients in medically underserved areas, such as rural areas, tend to have low health literacy [7]. Besides, older people cannot reach information effectively due to relatively low health literacy [7]. Furthermore, in rural areas, social norms may affect citizens’ perceptions of collaboration with others: many consider living without help a virtue, and this perception can be stronger among the older population [8]. Since higher levels of health literacy can facilitate effective health management and collaboration with healthcare professionals [9,10], improvement in health literacy can be expected to improve older patients’ level of health management, collaboration with healthcare professionals, and health outcomes. 

Social cognitive theory (SCT) is vital for clarifying rural citizens’ present conditions in health management. SCT can explain human behaviors based on three factors: individual, environmental, and behavioral factors [11]. In SCT, self-efficacy is one of the vital constructs that drive human behaviors. Individual factors consist of knowledge and outcome expectations, high levels of which can lead to high self-efficacy. Environmental factors include social support, opportunity/barrier, and social norms. By collaborating with others, moderating barriers, and changing social norms, the level of self-efficacy can increase. Behavioral factors consist of skills and accomplishment. Acquiring skills and the successive use of skills can lead to high levels of self-efficacy [11]. Driving the three factors can lead to the facilitation of health behaviors. This modeling significantly influences rural citizens’ participation in their health management [12]. Health interventions on participation in healthcare can drive rural citizens’ participation in their health management.

There is a need to develop educational interventions on participation in healthcare and evaluate the effects on citizens’ motivation to participate in their own healthcare [13]. We hypothesized that SCT-based educational interventions on participation in healthcare can improve older rural citizens’ self-efficacy in participating in their health management without difficulties. This field has received limited attention in the literature. Furthermore, there is a dearth of interventional studies on rural citizens’ perception of participation in health management. To fill this gap, in this study we investigate how the effectiveness of SCT-based educational interventions on participation in health management impacts rural citizens’ self-efficacy in participating in health management.

## 2. Materials and Methods

### 2.1. Participants

This study was conducted in Unnan, Shimane Prefecture, one of the most rural cities in Japan. In Unnan, a household typically constitutes of a nuclear family. Although Unnan City Hospital is the only general public hospital, the city has 16 clinics, 3 visiting-nurse stations, and 12 home care stations.

The participants were citizens of Unnan sampled from August 2018 to December 2018. They voluntarily communicated their intention to participate by consenting to attend an educational workshop held by Unnan City Hospital physicians. Unnan has 30 communities where citizens can select their preferred healthcare activities, such as receiving education on health promotion and disease prevention from medical care professionals and procuring support from the city hall [14]. Information on the educational workshop was publicized through the hospital’s newsletter and sent to each community in Unnan. Subsequently, three autonomous communities applied to participate in the workshop (total population of over-65-year-olds was 732). The control group participants were recruited from three communities near the hospital (total population of over-65-year-olds was 785), and the inclusion criteria included a willingness to participate and the ability to read and answer the questionnaire. In each community a community member provided the information regarding the education workshop to the citizens and recruited the participants in the community. The total number of participants was 277, including 156 in the intervention group and 121 in the control group.

### 2.2. Intervention

The educational workshop was designed based on SCT. The contents of the workshop were developed by RY and YR, who also consulted communities and asked for input from rural citizens during community meetings. The workshop was conducted in three communities, and its theme was: “How do you make decisions regarding your health conditions? A dialogue with medical professionals to help you answer this question”. The workshop was also based on knowledge–action theory [15] and provided participants with some knowledge and skills to improve their health management, considering the personal and behavioral factors of SCT [11,12]. The workshop content comprised general information on shared decision-making, health literacy, and collaboration with healthcare professionals for the improvement of the personal and environmental factors of SCT [11,12]. The workshop also included specific cases pertaining to chronic diseases. The explanation of cases included information on the importance of shared decision-making, health literacy, and collaboration with healthcare professionals, including the advantages of and difficulties encountered in shared decision-making and its method of execution, the relationship between health and health literacy, and the contribution of health literacy and shared decision-making. After the informative presentations, these concepts were put into practice. Regarding their own difficulties with healthcare, participants were encouraged to utilize their newly acquired knowledge to collaborate with healthcare professionals to solve their health problems. For the immediate application of the workshop content, participants were divided into groups of four to five members each and asked to take part in dialogue about their healthcare issues with other members and the physician. Finally, each group shared its dialogue content and received feedback from the physician and other participants. Through the workshop sessions, the importance of collaboration with healthcare professionals was continuously emphasized, as they may be reluctant to take part in dialogue with healthcare professionals to lead to a change in the social norms regarding healthcare. Each workshop, including the educational lecture, dialogue, and participants’ presentations, took approximately one and a half hours. 

### 2.3. Instruments

#### 2.3.1. Questionnaire

The questionnaire was developed exclusively for this study and included twelve items on patient participation; these were inspired by the Patient Preferences for Patient Participation tool (The 4Ps) [16] but phrased to match our study purpose. Further, while the 4Ps prescribes a parallel use of its two sections, including patient preferences and ex-periences, our questionnaire only captured citizens experiences of participation. (For more details and access to the 4Ps, see [16,17].) Further, our questionnaire asked study partici-pants to rate each item on a five-point scale from “1: I do not agree at all” to “5: I agree completely”. While the 4Ps have no subscales, we decided to consider headings, including (1) having a dialogue with healthcare staff, (2) sharing knowledge, (3) participating in plan-ning, and (4) managing self-care [17], in our questionnaire. The 4Ps has high validity and its original items are considered exhausting the concept of patient participation [16,17], yet were slightly revised to fit our study aims (Table 1). The questionnaire developed for this study was used to test changes in rural citizens’ perceptions of participation in health management and collaboration with healthcare professionals before and after the work-shops. We also collected data about participants’ age; sex (male/female); presence of a primary care doctor (yes/no); whether they made regular visits to their primary care phy-sicians (yes/no), having jobs (yes/no); and their self-rating on own health (good or rela-tively good = 1, relatively bad or bad = 0).

#### 2.3.2. Interview Guide

Interviews included the following questions: “What did you think of the workshop?” “What do you think of citizens’ participation in health management and collaboration with healthcare professionals?” “What are the advantages of participation in health management and collaboration with healthcare professionals?” “What are the disadvantages of participation in health management and collaboration with healthcare professionals?” “What are the roadblocks to participation in health management and collaboration with healthcare professionals?” The responses were qualitative in nature, and hence, were subjected to qualitative analysis.

#### 2.3.3. Procedure

The same survey questionnaire was administered before and one month after the educational workshops. The post-workshop questionnaire was sent to each community center’s clerk, who then distributed them among participants and collected the completed questionnaires within one week. Three communities that had not received the workshop were selected as the control group and given the questionnaire twice (before the beginning of the study and a month later) by volunteers in their respective communities.

Purposive sampling was used to select participants for the interviews based on whether they had participated in the workshop and had enough motivation to discuss their relationship with medical staff. Both male and female citizens and those with and without primary care physicians gave one-on-one semi-structured interviews that lasted approximately 20 min and were conducted by one of the two Unnan City Hospital physicians. These interviewers specialized in family medicine and home care and had experience in conducting interviews and qualitative research. They also presented on the issue of participation in health management and collaboration with healthcare professionals at the workshop. However, the interviewer was never the presenter of that particular workshop. Hence, while the workshop was conducted by one researcher, the participants were interviewed by the other, and vice versa. All the interviews were audio recorded and transcribed verbatim.

### 2.4. Analyses

To analyze the agreement between items and headings in our questionnaire among the intervention and control groups, a difference score was computed for each participant based on the pretest and posttest. These longitudinal difference scores were compared between the two groups using an unpaired t-test. Further, t-tests and chi-squared tests were used to test potential differences in background variables between the two groups. A significance level of *p* < 0.05 was used for all comparisons. This was determined so that a minimum of 63 participants were required in each group based on α (alpha) = 0.05, β (beta) = 0.20 (power of 80%). We used propensity score matching to adjust for the difference between the two groups’ backgrounds. To calculate the propensity scores, pre-workshop questionnaire results for age, sex, having a primary care physician, chronic diseases, employment, and self-rated health, were selected. One-to-one caliper (0.2) matching was used with no replacement for the matching of the intervention and control groups. The covariate balance between the matched groups was examined. All statistical analyses were performed using EZR (Saitama Medical Center, Jichi Medical University, Saitama, Japan), a graphical user interface for R (The R Foundation, Vienna, Austria) [18].

The qualitative data were thematically analyzed to extract themes and concepts related to citizens’ perceptions of their role in health management and collaboration with healthcare professionals. The analysis included becoming familiar with the data; generating initial codes; searching for themes; reviewing themes; defining and naming categories; and producing the report, including the selection of exemplification data and quotations. Transcripts were coded independently by R.O. and Y.R. and then checked for agreement during open coding. Subsequently, the researchers discussed the open codes and emerging concepts; further, they recoded or redefined concepts and categories wherever disagreements occurred. The data collection and analysis processes were conducted iteratively, and data collection continued until no new concepts emerged. To minimize personal bias, the transition from codes to preliminary themes and then to final themes included frequent discussions between the two authors. Further, to ensure member checking, the analysis was revealed to several participants, whose feedback was then included in the final revision of themes and concepts. No new themes emerged during the process of member checking.

### 2.5. Ethical Approval

Prior to conducting the study, participants were informed about the study’s aims and method of data disclosure and assured that their personal information would be protected, and their data would only be used for research purposes. Subsequently, the participants provided written consent to participate. Finally, this study was approved by the Unnan City Hospital clinical ethics committee (approval number: 20180032).

## 3. Results

### 3.1. Demographic Data

In the intervention group, the response rates of the pretest and posttest were 100% and 78.2%, respectively. In the control group, the response rate was 98.3% in both categories. Participants in the intervention group were older than those in the control group (*p* < 0.001). No significant differences in conditions regarding having a primary care physician, gender, employment, and self-rated health were found between the groups. After adjusting for propensity score matching, 71 participants of the intervention group were matched with 71 participants in the control group. The C statistic for propensity score regression models was 0.756 (95% confidence interval (CI), 0.694–0.818) (Table 2).

### 3.2. Difference in Citizens’ Perceptions of Participation in Health Management and Collaboration with Healthcare Professionals between the Intervention and Control Groups

After propensity score matching, the scores for the subscales “having a dialogue with healthcare staff” (*p* = 0.696) and “sharing knowledge” (*p* = 0.217) were not significantly different between the intervention and control groups (Table 2). For the subscales “partaking in planning” (*p* = 0.017) and “managing self-care” (*p* = 0.003), the scores of the posttest were greater in the intervention group than in the control group (Table 3).

### 3.3. Difficulties Faced by Rural Citizens Participating in Health Management and Collaborating with Healthcare Professionals

Interviews were conducted for 17 participants, among whom the average age was 70.4 years (SD = 3.9), and nine were men (52.3%). Through direct content analysis, three themes and seven concepts were described. The three themes were ability to manage health conditions, relationship with medical professionals, and relationship among citizens (see Table 4 for more details on themes and concepts).

### 3.4. Ability to Manage Health Conditions

#### 3.4.1. Difficulty in Judging Symptoms

Rural citizens come from various, mostly nonmedical, backgrounds. On presenting a symptom, they may not be able to determine whether they should visit a medical institution. If their relatives or neighbors have adequate medical knowledge, they might advise them to seek appropriate medical attention. However, this is not always the case. Rural citizens’ attitudes toward physicians depend on their relationships with the latter and the resources available in their communities. Often, they are motivated to participate in health management and collaborate with healthcare professionals; however, they hesitate to collaborate among themselves since they lack appropriate medical knowledge.

#### 3.4.2. Limited Collaborative Experience

The workshop enabled participants to understand the importance of participation in health management and collaboration with healthcare professionals; however, the subjects had never experienced its effectiveness and hence were unable to understand the work done by each medical professional. Further, the citizens found it difficult to imagine collaborating with medical professionals. Generally, rural citizens tend to passively experience medical situations, which prohibits them from considering their role in health management and collaboration with healthcare professionals.

### 3.5. Relationship with Medical Professionals

#### 3.5.1. Hierarchy in Healthcare

Rural citizens perceive a hierarchical structure that separates them from medical professionals, particularly doctors. Traditionally in Japan, paternalism has prevailed in health services, and patients are expected to accept physicians’ decisions regardless of their own preferences. Today, although the shared decision-making process is becoming popular, rural citizens continue to conform to paternalism. Hence, despite feeling anxious about their treatment and preferences, they are unable to clearly state their wishes in face-to-face dialogues.

#### 3.5.2. Feelings of Low Self-Efficacy

Despite understanding the importance of participation and collaboration in improving their health and that of their families, some rural citizens feel inadequate while participating in health management and collaborating with healthcare professionals. They have low self-efficacy in terms of participation in health management and collaboration with healthcare professionals and require more exposure to similar interventions. 

### 3.6. Relationships among Citizens

#### 3.6.1. Weakening Relationships among Citizens

Today, rural communities are changing—young people are migrating from these areas, and the population left behind is growing old. These conditions are causing changes in the social structure and reducing the number of interactions among rural citizens. Rural citizens often feel unable to sufficiently help each other, which makes them anxious about their future and hinders their participation in health management and collaboration with healthcare professionals. Hence, gradual changes in social conditions and perceptions of participation in health management and collaboration with healthcare professionals may negatively affect citizens’ lives.

#### 3.6.2. Anxiety about Privacy

Rural citizens feel anxious about privacy. In the past, they were quick to share information among themselves. However, in recent times, issues associated with the maintenance of privacy have become prevalent, and public disclosure of personal information and the spread of incorrect personal information have created significant problems.

#### 3.6.3. Cultural Norms

Often, rural citizens believe that they should live on their own and are proud of their independence. Despite knowing that medical professionals and neighbors are willing to help, citizens are reluctant to ask. Therefore, social norms and cultivated emotions may prevent rural citizens from asking for and receiving competent help.

## 4. Discussion

The SCT-based educational intervention partially improved the rural citizens’ perceptions of participation in health management and collaboration with healthcare professionals, particularly regarding partaking in planning and managing self-care. Regarding the existence of dialogue with healthcare staff and sharing knowledge, there was no statistically significant difference between the two groups. The high scores of both groups regarding the latter two aspects may imply that rural citizens are already motivated to talk to healthcare staff and share their conditions, at levels that require less patient empowerment than needed for shared-decision making. Moreover, our findings demonstrate that rural citizens can be motivated to participate in health management, collaborate with healthcare professionals, and engage in shared decision-making with medical professionals, if they are supported by offering them the required education. To promote rural citizens’ activities in managing their healthcare and prevent diseases, they should be empowered to participate in their healthcare [19,20,21]. Based on the results of this study, which contradict those of earlier studies in other countries and urban areas [8], rural citizens can be easily motivated to participate in health management and collaborate with healthcare professionals, using their long-established relationship with their primary care physicians. However, their current inability to take appropriate action to realize the importance of participation in health management and collaboration with healthcare professionals necessitates investigation into citizens’ beliefs regarding intent for activities ensuring participation in health management and collaboration with healthcare professionals.

Rural citizens perceive difficulties in contacting appropriate medical professionals at the appropriate time. Primary care physicians should ensure that citizens contact the right medical professionals [22]. Further, primary care physicians should provide various types of support to improve citizens’ knowledge of their symptoms, which may improve the citizens’ attitudes toward taking action when first experiencing symptoms [23,24]. Although they often lack sufficient medical knowledge to judge their symptoms, simple notifications on dangerous symptoms and clinical courses may help them understand when they should visit medical institutions [25]. This information should be provided by primary facilities; however, if there are only a few such facilities in rural areas, hospitals should disseminate this information [26].

Citizens perceive a hierarchy between medical professionals and themselves, and hence, are prone to feelings of low self-efficacy in healthcare. Traditionally, paternalism has been the dominant aspect of the relationship between medical professionals and patients [27,28]. In rural areas, such traditions persist in various professions, including medicine [29]. Rural citizens highly respect their primary care physicians and are, in general, reluctant to express their healthcare intentions and preferences to their doctors, which often leads to a lack of motivation to participate in health management and collaborate with healthcare professionals [30,31]. Moreover, this lack of motivation and opportunities to participate in health management and collaborate with healthcare professionals may cause a further decline in the citizens’ self-efficacy [32,33]. 

Today, Japan is experiencing changes in its social structure; accordingly, communities may be forced to modify their social conditions, for example, by promoting citizens’ participation in health management and their collaboration with healthcare professionals. Young people are leaving rural areas and the population left behind is growing older. Furthermore, people in rural areas meet each other less frequently today than in the past [27]. This change may negatively affect the relationship among citizens and prevent them from providing mutual assistance. Furthermore, their cultural values make them reluctant to depend on each other, although mutual assistance during emergencies was an essential aspect of rural life in the past. Hence, both the change in relationships and the persistent traditional characteristics of rural people may prevent them from helping each other [34]. However, these social changes can be prevented; the provision of adequate information and education and assigning a community coordinator may help rural communities overcome their health issues [35]. Going forward, health care education interventions only through rural physicians might be unrealistic because there are only a few physicians in rural settings. Hence, we also recommend the training of other healthcare professionals as instructors for this workshop, and even of rural citizens who are motivated to further educate their communities. Future studies may investigate and facilitate the spread of education on the usage of healthcare and collaboration with healthcare professionals to large communities of rural citizens.

The current study has some limitations. First, although its results are applicable to rural communities in Japan, they may be incompatible with rural settings in other parts of the world. However, all rural areas have strong social norms that affect their citizens’ relationships with medical professionals; therefore, this study may be referenced by researchers conducting interventions in other rural areas. The second limitation is the nonrandomization in sampling and the use of a non-validated questionnaire in Japanese. In this pilot study, through the discussion with citizens and the local government about the educational workshop, we chose a non-randomized method owing to the lack of resources and in order to respect the ethical issues of citizens’ perception of unfairness in medical care in the control groups. For the adjustment of background difference, we used the propensity score matching. Further studies may be performed with randomization and validated questionnaires. The third limitation is the relationship between interviewers and interviewees. Although there were no strong relationships between them, the participants might still have felt some pressure in the interviews. That is, as the relationship between patients and physicians tends to be hierarchical, the participants might not have been able to easily share their opinions or feelings about their physicians. 

## 5. Conclusions

This study clarified that SCT-based educational intervention can change rural citizens’ perceptions of participation in health management and collaboration with healthcare professionals, particularly regarding self-care and the care provided by healthcare professionals. Although rural citizens want to participate in healthcare activities, they do not know how to do so. Educational interventions could be useful in disseminating knowledge of healthcare and participation in health management. By teaching citizens to appropriately judge symptoms and construct collaborative systems, healthcare activities in communities can be supported.

## Figures and Tables

**Table 1 ijerph-18-01782-t001:** The items conceptualizing citizen participation used in this study.

No	Heading	Revised Items
1	Having dialogue with health-care staff	Having good conditions for mutual communication with health-care staff is essential.
2		My knowledge and preferences regarding my care should be respected.
3		Health-care staff should listen to you.
4	Sharing knowledge	I want to get explanations for my symptoms/issues.
5		I would like to tell about my symptoms/issues.
6		Health-care staff should explain to me the procedures to be performed/that are performed
7	Partaking in planning	I should know what is planned for me.
8		I would like to take part in the planning of my care and treatment
9		I want to be able to phrase personal goals.
10	Managing self-care	I would like to perform self-care e.g., managing my medication or changing dressing.
11		I prefer to manage self-care e.g., adjusting diet or performing preventive health care.
12		I would like to know how to manage my symptoms.

**Table 2 ijerph-18-01782-t002:** Demographic data of participants in each group, and the significance level of the comparison among the two groups before and after propensity score matching.

	Crude			After Propensity Score Weighting		
Variables	Intervention (*n* = 122)	Control (*n* = 119)	*p*-Value	Intervention (*n* = 71)	Control (*n* = 71)	*p*-Value
Gender, (male: N, %)	57 (46.7)	60 (50.4)	0.607	33 (46.5)	34 (47.9)	1
Age in years (M, SD)	71.38 (3.38)	68.58 (3.14)	<0.001	69.58 (2.89)	69.89 (3.24)	0.549
Primary care physician (N_yes_, %) ^1^	89 (72.9)	89 (73.6)	0.156	55 (77.5)	53 (74.6)	0.846
Chronic diseases (N_yes_, %) ^2^	98 (80.1)	88 (73.9)	0.283	54 (76.1)	51 (71.8)	0.703
Employment (N_yes,_ %) ^3^	85 (69.7)	89 (74.8)	0.473	51 (71.8)	55 (77.5)	0.563
Self-rated health (N_high_, %) ^4^	89 (73.0)	91 (76.5)	0.392	51 (71.8)	54 (76.1)	0.703
Headings in study questionnaire						
Having dialogue with healthcare staff (SD) ^5^	12.55 (1.37)	12.83 (1.30)	0.116	12.75 (1.33)	12.75 (1.32)	1
Sharing knowledge (SD)	12.94 (1.07)	13.17 (1.40)	0.161	13.06 (1.13)	13.04 (1.53)	0.95
Partaking in planning (SD)	12.17 (1.40)	12.10 (1.48)	0.706	12.28 (1.28)	12.01 (1.41)	0.238
Managing self-care (SD)	12.05 (0.93)	12.02 (1.55)	0.845	12.13 (0.97)	11.82 (1.46)	0.138

Note. ^1^ Primary care physician: Do you have a primary care physician?; ^2^ Chronic diseases: Do you regularly visit your primary care physician for your diseases?; ^3^ Employment: Do you have a job?; ^4^ Self-rated health: Do you consider yourself healthy?; ^5^ SD: Standard deviation.

**Table 3 ijerph-18-01782-t003:** Longitudinal difference scores (changes between pretest and posttest) on the four headings of the questionnaire: Description and results of a comparison of groups before and after propensity score matching.

Variables	Crude			After Propensity Score Weighting		
	Intervention (*n* = 122)	Control (*n* = 119)	*p*-Value	Intervention (*n* = 71)	Control (*n* = 71)	*p*-Value
Having dialogue with healthcare staff (SD) ^1^	12.68 (1.12)	12.83 (1.34)	0.342	12.80 (1.23)	12.76 (1.34)	0.845
Sharing knowledge (SD)	13.18 (1.19)	13.13 (1.41)	0.784	13.30 (1.24)	13.00 (1.50)	0.202
Partaking in planning (SD)	12.39 (1.02)	12.14 (1.41)	0.115	12.62 (0.88)	12.04 (1.35)	0.003
Managing self-care (SD)	12.43 (1.04)	11.96 (1.60)	0.007	12.49 (1.01)	11.76 (1.53)	0.001

^1^ SD: Standard deviation.

**Table 4 ijerph-18-01782-t004:** Themes and concepts developed through direct content analysis.

Theme	Concept	Quotes
Ability to manage health conditions	Difficulty in judging symptoms	Through this workshop, I could confirm the importance of us participating in health management and collaborating with healthcare professionals. However, although I know that we should visit medical institutions if we have medical problems, to get healthy, we cannot determine the seriousness of our symptoms. We find medical issues confusing. (Interviewee B)
	Limited collaborative experience	This was our first time understanding the importance of health management and collaboration with healthcare professionals. I understand that we should participate in the collaboration. However, overall, we do not know each professional’s work and how they collaborate between themselves. I want to learn more about each profession. (Interviewee K)
Relationship with medical professionals	Hierarchy in healthcare	We are used to following a physician’s decisions. When it comes to treatment, I cannot imagine a situation in which I can provide my opinion on such decisions. Also, we do not have appropriate medical knowledge, and following their choices may be safer for us. (Interviewee M)
	Feelings of low self-efficacy	We understand the importance of health management and collaboration with healthcare professionals and our participation in it. I am interested in it, but I do not feel any confidence while doing it. It may be the result of our lack of experience or limited understanding of the collaboration. I cannot say correctly. We may need more education in this regard. (Interviewee O)
Relationship among citizens	Weakening connections among citizens	Times have changed. In the past, we could help each other when we were in trouble. Especially when we had certain symptoms, some neighbors consulted medical professionals for us, which enabled early diagnosis and better treatments. However, in the present community, the loose connections between citizens prevent such collaborations, which may lead to the delayed treatment of critical diseases and mortality. (Interviewee K)
	Anxiety about privacy	Now, we are very anxious about privacy and cannot share information easily, even in rural communities. Of course, privacy is essential if we want everyone to be secure in their homes. However, too much is dangerous, especially in rural areas where there are only limited resources. To collaborate with medical professionals, we must balance our need for privacy and safety. (Interviewee G)
	Cultural norms	We may need help in reality, but we may not feel it. Everybody must think so, when they manage to live by themselves, and in the later stages of their lives, they have to ask for some help. This trend may be strong in rural areas. (Interviewee C)

## Data Availability

The datasets used and/or analyzed during the current study are available from the corresponding author upon reasonable request.

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
