# Peer review of "Educational Intervention to Improve Citizen’s Healthcare Participation Perception in Rural Japanese Communities: A Pilot Study"

_ijerph, 2021, doi:10.3390/ijerph18041782_

Round 1
Reviewer 1 Report
This is an interesting study about an educational workshop for older rural persons enhancing health literacy. Overall, this is not my field of expertise, so I sometimes found it hard to comprehensively read the article. However,there are some aspects I want to adress:
I would suggest to focus on the qualitative aspects of the study and I would also probably include these phrases in the title (qualitative, pilot study).
Introduction: The introduction is lacking the information why this kind of training is especially important for elderly citizens or why the authors decided to include the older population only.
Materials and Methods:
- I would recommend to include more informations and details on the workshop itself, how it was developed, who developed the intervention, what exactly is the content of the workshop and how much time is spent on each content. How many participants are planned in each workshop, how many workshops are planned?
- Why did the authors did not plan an RCT?
- The first paragraph in the participants section (p2 line 72ff) is exactly the same as the last one of the introduction (p2 line 63ff), I would delete the second one.
- In line 88 the authors say "four communities", but in line 95 they say "five communities" - what is correct?
- How did the participants get the information about the intervention?
- Could the authors describe the four scales of the questionnaire in more detail? I think this could also be helpful for the discussion (see recommendations for discussion)
Instruments:
The authors describe that they used a questionnaire that was adapted and translated. So the questionnaire has not been evaluated and has even been changes, this is why I would strongly reccomend to focus on the qualitative aspects.
Line 134: how is the "self-rated health" rated?
line 138: What do the authors mean by "What did you think of the workshop this time?" - what is meant by the phrase "this time"? Do the participants get workshops more than once?
Results: line 204: Participants of the intervention group were older than in the control group, what does that mean? Is this a limitation? Authors should discuss that point somehow as is is the only baseline difference.
Discussion:
first line of discussion: I would insert "SCT based educational intervention partly improved...", as only two of four scales improved if I got that right?
I would further recomment that the authors discuss more in detail why they think that the two scores improved and the others not. What does that mean for clinical significance? What do they think might help to also improve the other two scales? And how can partaking in planning be enhanced without having a dialogue with healthcare staff? (thats why I would appreciate to get more information regarding the scales of the questionnaire).
Limitations: To me, the core limitation is the lack of evaluated and validated instruments, this should be mentioned here. Another limitation is the rural area and the age included group and that the study is no RCT.
Author Response
This is an interesting study about an educational workshop for older rural persons enhancing health literacy. Overall, this is not my field of expertise, so I sometimes found it hard to comprehensively read the article. However,there are some aspects I want to adress:
I would suggest to focus on the qualitative aspects of the study and I would also probably include these phrases in the title (qualitative, pilot study).
Response:
We thank the reviewer for this insightful comment. In response to this comment, we have added the phrase “a pilot study” in the title. Although we have not included the term “qualitative” in the title, we used your suggestion to alter the title to shift its focus from the study design to the actual aspect being studied. We have emphasized the qualitative nature of the study in the methods section.
Introduction: The introduction is lacking the information why this kind of training is especially important for elderly citizens or why the authors decided to include the older population only.
Response:
We thank the reviewer for this insightful comment. We have now revised the description of the training in the background section, adding the rationale behind focusing on rural older population (see lines 38-45).
Materials and Methods:
- I would recommend to include more informations and details on the workshop itself, how it was developed, who developed the intervention, what exactly is the content of the workshop and how much time is spent on each content. How many participants are planned in each workshop, how many workshops are planned?
Response:
We thank the reviewer for this insightful comment. We agree with the suggestion. In response to this comment, we have revised the section of participants and intervention by adding the details of the participants and their selection process, and details of the intervention (lines 81-100).
- Why did the authors did not plan an RCT?
Response:
In response to this comment, we have revised the section on limitations by adding the reason of not planning an RCT (lines XX-YY).
- The first paragraph in the participants section (p2 line 72ff) is exactly the same as the last one of the introduction (p2 line 63ff), I would delete the second one.
Response:
We thank the reviewer for pointing this out. We have deleted the indicated part.
- In line 88 the authors say "four communities", but in line 95 they say "five communities" - what is correct?
Response:
We thank the reviewer for pointing this out. We have now clarified the number of communities participating in the study in the sections Participants and Intervention.
- How did the participants get the information about the intervention?
Response:
We thank the reviewer for this insightful comment. We agree that this information should be included in the paper. In response to this comment, we have added the content regarding communicating the information of the intervention. This is added in the Participants section (lines 70-79).
- Could the authors describe the four scales of the questionnaire in more detail? I think this could also be helpful for the discussion (see recommendations for discussion)
Response:
In response to this comment, we have elaborated the description of the questionnaire by adding more information about the 4Ps tool and on how we adapted items for this study. We also added a table with the items per subscale.
Instruments:
- The authors describe that they used a questionnaire that was adapted and translated. So the questionnaire has not been evaluated and has even been changes, this is why I would strongly reccomend to focus on the qualitative aspects.
Response:
We thank the reviewer for this insightful comment. We have revised the description of the questionnaire by adding the detailed information and showing the minor adaptations made in Table 1. We do not agree on the need to exclude these data from the paper, but we have emphasized the qualitative analysis of interviews and indicated the limitation of this study regarding the use of this questionnaire and the recommendation to use a validated instrument in future research in the discussion part.
- Line 134: how is the "self-rated health" rated?
Response:
We have revised the survey questionnaire section and added information of rating scale used for self-rated health (lines 108-116).
- line 138: What do the authors mean by "What did you think of the workshop this time?" - what is meant by the phrase "this time"? Do the participants get workshops more than once?
Response:
Thank you. In response to this comment, we have revised the description of the interview guide and deleted the mistranslated part.
- Results: line 204: Participants of the intervention group were older than in the control group, what does that mean? Is this a limitation? Authors should discuss that point somehow as is is the only baseline difference.
Response:
We thank the reviewer for pointing this out. We agree with the suggestion. In response to this comment, we have revised the methods part including the explanation of the propensity score matching for the adjustment of the difference in the background data (lines 148-151).
Discussion:
- first line of discussion: I would insert "SCT based educational intervention partly ..", as only two of four scales improved if I got that right?
Response:
We thank the reviewer for this insightful comment. We agree with the suggestion. In response to this comment, we have inserted the word “partially.”
- I would further recomment that the authors discuss more in detail why they think that the two scores improved and the others not. What does that mean for clinical significance? What do they think might help to also improve the other two scales? And how can partaking in planning be enhanced without having a dialogue with healthcare staff? (thats why I would appreciate to get more information regarding the scales of the questionnaire).
Response:
We thank the reviewer for this suggestion and agree to it. In response, we have revised the Discussion regarding the reason why the two categories showed improvement but the others did not.
- Limitations: To me, the core limitation is the lack of evaluated and validated instruments, this should be mentioned here. Another limitation is the rural area and the age included group and that the study is no RCT.
Response:
We agree with the reviewer’s viewpoint. In response to this comment, we have revised the limitation by adding the reason of not planning an RCT and the lack of an evaluated and validated instrument.
Reviewer 2 Report
Dear authors
A very interesting research design on an important topic, and very well done overall.
I just have a couple of concerns that you might remedy:
- Section 3.2 you give the quantitative results but do not discuss why some changes pre- to post test are significant and why some are not. This is also a very short section. It would be interesting to hear why you think two of the measures had significance and two did not, by discussing each in turn.
- The discussion of limitations in the conclusion: how "real world" is the workshop intervention? Surely it is not realistic to think that doctors will run workshops for everyone in a rural area for 1.5 hours to try and get them more comfortable with taking an active role in their healthcare management. Thus, what real-world conclusions can we draw here? Perhaps some, but the significance would lessen as the workshops turned into pamphlets, commercials, public messaging, or shorter interventions more likely to be affordable by the state. Should discuss this issue at much more length.
Author Response
A very interesting research design on an important topic, and very well done overall.
I just have a couple of concerns that you might remedy:
- Section 3.2 you give the quantitative results but do not discuss why some changes pre- to post test are significant and why some are not. This is also a very short section. It would be interesting to hear why you think two of the measures had significance and two did not, by discussing each in turn.
Response:
We thank the reviewer for appreciating the study and for this insightful comment. In response to this comment, we have revised the first paragraph of the Discussion regarding the reasons why the two categories showed improvement but the others did not.
- The discussion of limitations in the conclusion: how "real world" is the workshop intervention? Surely it is not realistic to think that doctors will run workshops for everyone in a rural area for 1.5 hours to try and get them more comfortable with taking an active role in their healthcare management. Thus, what real-world conclusions can we draw here? Perhaps some, but the significance would lessen as the workshops turned into pamphlets, commercials, public messaging, or shorter interventions more likely to be affordable by the state. Should discuss this issue at much more length.
Response:
We thank the reviewer for this insightful comment. We agree with the point and added a comment and suggestion regarding the involvement of other healthcare professionals and citizens in communities in the Discussion.
Reviewer 3 Report
The manuscript entitled “Educational intervention to improve citizen’s healthcare participation perception in rural Japanese communities” presents an interesting issue associated with educational interventions.
Abstract:
- Line 17 – “Interview contents were thematically analyzed” – it should be explained more detail
Materials and Methods
- Lines 72-76 – “This field has received limited attention in the literature. Furthermore, there is a dearth of interventional studies on rural citizens’ perception of participation in health management. To fill this gap, in this study, we investigate how the effectiveness of SCT-based educational interventions on participation in health management impacts rural citizens’ self-efficacy to participate in health management.” – this pragraph should be removed.
- inclusion and exclusion criteria for participant should be presented.
- 4Ps tool - what is the original language of the questionnaire? Original language English? Was the questionnaire translated (translation and back-translation)? Who did so? Any validation of the translated questionnaire?
- More information is needed about the validity and reliability of each measure. This is a crucial issue. Additionally, any limitations in reliability and validity need to be addressed in the discussion.
Results and discussion
- The international context should be presented in this section – for international readers. Authors should compare gathered data with the results by other authors,
- Some additional information should be added to the limitations sections (e.g. Participants in the intervention group were older than those in the control group).
Author Response
The manuscript entitled “Educational intervention to improve citizen’s healthcare participation perception in rural Japanese communities” presents an interesting issue associated with educational interventions.
Abstract:
- Line 17 – “Interview contents were thematically analyzed” – it should be explained more detail
Response:
We have now reformulated this sentence and added the focus of the thematic analysis.
Materials and Methods
- Lines 72-76 – “This field has received limited attention in the literature. Furthermore, there is a dearth of interventional studies on rural citizens’ perception of participation in health management. To fill this gap, in this study, we investigate how the effectiveness of SCT-based educational interventions on participation in health management impacts rural citizens’ self-efficacy to participate in health management.” – this pragraph should be removed.
Response:
In response to this comment, we have deleted the suggested part.
- inclusion and exclusion criteria for participant should be presented.
Response:
We thank the reviewer for this comment and agree with the suggestion. In response to this comment, we have added the criteria in the participants section.
- 4Ps tool - what is the original language of the questionnaire? Original language English? Was the questionnaire translated (translation and back-translation)? Who did so? Any validation of the translated questionnaire?
Response:
We thank the reviewer for this comment. We have revised the description of the questionnaire by adding the requested information.
- More information is needed about the validity and reliability of each measure. This is a crucial issue. Additionally, any limitations in reliability and validity need to be addressed in the discussion.
Response:
In response to this comment, we have elaborated the description of the instrument in the Method. Additionally, in the limitation section in the Discussion, we mention the methodological limitations of the instrument used and addressed the need for validated instruments in future research.
Results and discussion
- The international context should be presented in this section – for international readers. Authors should compare gathered data with the results by other authors,
Response:
We have now added the comparison of our results with findings from other settings/countries. We made this addition in the Discussion section.
- Some additional information should be added to the limitations sections (e.g. Participants in the intervention group were older than those in the control group).
Response:
In response to this comment, we have revised the Methods part by including the explanation of the propensity score matching for the adjustment of the difference of the background data. Additionally, in the Discussion we describe the limitation that this research was a non-randomized study.
Round 2
Reviewer 3 Report
I have no further comments. I would like to congratulate you an interesting research.